

# Descriptive profile of risk factors for cardiovascular diseases using WHO STEP wise approach in Madhya Pradesh

Arun M. Kokane[1], Rajnish Joshi[2], Ashwin Kotnis[3], Anirban Chatterjee[1], Kriti Yadav[1], G Revadi[1], Ankur Joshi[1] and Abhijit P. Pakhare[1]

[1] Community and Family Medicine, All India Institute of Medical Sciences, Bhopal, Bhopal, Madhya Pradesh, India
[2] General Medicine, All India Institute of Medical Sciences, Bhopal, Bhopal, Madhya Pradesh, India
[3] Biochemistry, All India Institute of Medical Sciences, Bhopal, Bhopal, Madhya Pradesh, India

## ABSTRACT

**Background.** Periodic information on risk factor distribution is critical for public health response for reduction in non-communicable disease (NCDs). For this purpose, the WHO has developed STEPs wise approach. State representative population-based STEPS survey was last conducted in 2007–08 in seven states of In India. Since then no such work has been reported from low ETL states. This survey was carried out to assess the prevalence of risk factors associated with NCDs and the prevalence of NCDs in the low ETL state of Madhya Pradesh using the WHO STEPs approach.

**Methods.** A total of 5680 persons aged 18–69 years were selected from the state of Madhya Pradesh using multi-stage cluster random sampling. Using the WHO STEPs approach, details were collected on demographics, STEP 1 variables (tobacco consumption, alcohol consumption, physical activity, diet), STEP 2 variables (weight, height, waist circumference, blood pressure) and STEP 3 variables (fasting blood glucose, blood cholesterol).

**Results.** We found that 9.4% individuals smoked tobacco, 15.3% were overweight/obese, 22.3% had hypertension, and 6.8% have diabetes mellitus. As compared to women, men were less likely to be overweight or obese, but more likely to smoke tobacco, and have diabetes mellitus. Hypertension was also more common in men. Overall, about a fourth of all adults had three or more risk factors for cardiovascular disease.

**Conclusion.** The survey shows that a large section of the population from Madhya Pradesh is either suffering from NCDs or have risk factors which predispose them to acquire NCDs. This state representative survey provides benchmarking information for behavioural and biological risk factor distribution for recently scaled up National Programme for the Prevention and Control of Cancer, Diabetes, Cardiovascular Diseases, and Stroke (NPCDCS).

Corresponding author
Abhijit P. Pakhare,
abhijit.cfm@aiimsbhopal.edu.in

## INTRODUCTION

Various states of India are passing through an epidemiological transition. This has resulted in an increase in the burden of non-communicable diseases while being saddled with a high burden of communicable diseases at the same time. The Global Burden of Disease (GBD) 2017 study found that for the year 2017, non-communicable diseases (NCDs) accounted for 6.2 million deaths in India alone (*Institute of Health Metrics and Evaluation, 2017*). India already has largest number of individuals living with diabetes, most of which is related to being overweight or obese (*Joshi & Parikh, 2007*). We also have a high prevalence of hypertension, which is estimated to be around 30 to 40% in all adults (*Ramakrishnan et al., 2019*; *Tripathy et al., 2017*). Being second most populous country in the world, we soon will be a dual hypertension as well as diabetes capital of the world. Although initially it was seen that NCDs are primarily a disease of the affluent, ongoing studies have subsequently shown that NCDs are common in the poor and vulnerable sections of the society (*Menon et al., 2015*).

Various local surveys have highlighted the magnitude of NCD problem in India over the years. The WHO STEPS approach was designed in order to strengthen these initiatives, and to standardize the survey methodologies (*World Health Organization, 2017*). Subsequently, various local surveys using WHO STEPS methodology have reported on prevalence of NCD risk-factors from various parts of the country (*Ministry of Health of Vietnam, 2016*; *Kenya National Bureau of Statistics, 2015*; *World Health Organization (WHO), 2010*; *Nepal Health Research Council, 2013*; *National Institute of Medical Statistics, 2007*; *Bhar, Bhattacherjee & Das, 2019*; *Bhardwaj et al., 2014*; *Bhattacherjee et al., 2015*; *Shikha et al., 2019*; *Singh et al., 2017*). Given that there is a high chance of a wide heterogeneity in estimates from these local surveys, need for State-wide surveys and even a national survey was felt. State wide surveys from Indian states of Punjab, Kerala and Haryana have been reported in last few years (*Thakur et al., 2016*; *Thakur et al., 2019*; *Sarma et al., 2019*). A national NCD survey was completed in year 2018–19, and its results are currently awaited. Since programmatic interventions in NCDs need to be tailored to needs of the States, such surveys would be instrumental in defining the public health magnitude and consequently help in designing control strategies for the same.

Almost all NCDs are associated with modifiable behaviours. Tobacco consumption, physical inactivity, harmful use of alcohol and the increasing popularity of unhealthy dietary habits are all responsible for increasing the risk, and therefore the incidence of NCDs (*World Health Organization, 2019*). Multiple studies conducted with and without the use of the STEPS tool have shown variation in prevalence of behavioural risk factors across the country over a period of time. For example, the consumption of tobacco related products is declining across the country, but at the same time the number of people living a sedentary lifestyle is increasing (*Devamani et al., 2019*; *Newtonraj et al., 2017*; *Anjana et al., 2014*; *Nethan, Sinha & Mehrotra, 2017*), and so is alcohol consumption (*Prasad, 2009*). Since periodic national surveys will always be a challenge, methodical periodic state wide surveys would be useful to monitor the secular trends of behavioural risk factors and establish a true surveillance system for NCDs. Such data obtained through periodic

surveillance of risk factors is necessary for public health policy makers to design and dynamically adapt public health programs directed towards reduction of NCDs in the community.

India has 28 states and eight union territories, Madhya Pradesh (MP) being second largest in area, with the highest proportion of tribal population from among all states in the country (*ST, 2020*). It is also one fraught with socioeconomic challenges, and is therefore classified as an empowered action group (EAG) state. All EAG states are classified at a low epidemiological transition level (ETL) (*Dandona et al., 2017*). ETL is based on a ratio of Disability Adjusted Life Years (DALYs) from communicable, maternal, neonatal, and nutritional diseases (CMNNDs) to those from non-communicable diseases (NCDs) and injuries. States in the Low ETL group (CMNND/NCD mortality ratio between 0.56 and 0.75) and are yet to see a huge rise in NCD mortality unlike high ETL states of Kerala, and Punjab (ratio less than 0.31). Yet the sheer number of people living in these states makes it necessary to unearth the burden of NCDs in these states, so as to guide development of public health policies according to the local context. Previous state-wide surveys are all from high ETL states, and the current study is first from a low ETL state.

## METHODS

### Study design and settings

This was a community based cross-sectional survey conducted in the state of Madhya Pradesh (MP) located in central part of India. Madhya Pradesh state has 51 districts spread across 10 administrative divisions. As per 2011 census, total population of the state was 72.6 million of which 15.34 million (21.1%) belong to *Scheduled Tribes*. MP is afforded the label of being one of the high priority states under National Health Mission considering its high infant mortality rate (IMR), maternal mortality rate (MMR) and high incidence of malnutrition in children. The present survey was conducted in 10 districts of MP during February 2018 to March 2019.

### Sample size and sampling method

We have used multi-stage cluster random sampling method as recommended in WHO-STEPS wise approach. Sample size calculations were done by using STEPS Sample Size Calculator (*World Health Organization (WHO), 2020a*) for estimating prevalence of any form of tobacco use with 95% confidence interval around relative error of 10% and adjusted for design effect of 1.5, and also for four age-group strata (18–29, 30–44, 45–59 and 60–69 years) for both gender and non-response rate of 20%. Calculated sample size was 5741. Considering logistics feasibility we have decided our cluster size to be 60 participants per cluster and thus a total of 100 clusters were selected.

In the first stage, all divisions from within the state were selected and in second stage one district from each division was randomly selected. Figure 1 shows map of Madhya Pradesh and selected clusters and districts. We have used random number table and census code to identify the districts included in our study.

Then in third stage, sampling frame was prepared by using existing enumeration areas i.e., Primary Sampling Units (PSU) of the Census 2011 and from this unit, sample was

**SELECTED DISTRICTS IN MADHYA PRADESH STEPS SURVEY 2017-2018**

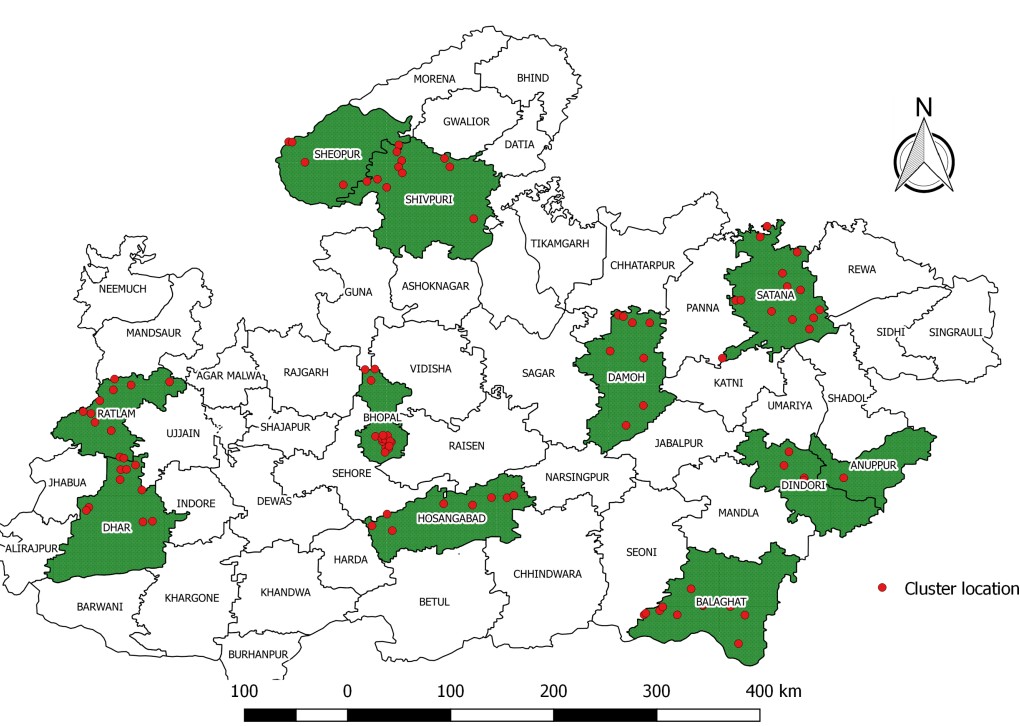

**Figure 1** **Selected clusters and Districts for NCD STEPS Survey.** Dot indicate location of cluster and shaded area indicates selected district.

chosen. All the villages / Census enumeration blocks (CEB) in wards of the selected districts of Madhya Pradesh were listed as Primary Sampling Unit (PSU). A total of 100 PSUs were selected by proportionate to population size of district considering urban and rural stratification. Total number of urban and rural clusters selected are shown in Table 1.

Then from each district, corresponding number of clusters/PSU were selected by probability proportionate to size (PPS) method. For this we prepared list of villages and CEBs with their population size. Cumulative population of the enlisted villages and CEBs was calculated and it was divided by the requisite number of clusters to arrive at the sampling interval. First cluster was chosen by selecting a random number between 1 to sampling interval. Subsequent clusters were selected by adding sampling interval.

From each cluster we approached 58 eligible individuals for participation in study. For this, field investigators did village street mapping and house listing and then divided total number of households in that cluster by 58. This was the sampling interval for household selection. For selection of first household, a street was randomly selected by using last digit of currency note, and then first household adjacent to that street was selected. Then all eligible individuals from selected household were enlisted and assigned a serial number in paperless data collection tool. This tool had inbuilt logic based on Kish grid method to select a respondent from eligible household members. Subsequently, households on right hand side of the household being interviewed currently were selected by adding sampling

**Table 1  Selected districts and number of cluster per district.**

| District (Census Code) | Total persons | No. of villages | No. of CEBs in urban areas | Total clusters to be selected | Rural clusters | Urban clusters | Tribal population % |
|---|---|---|---|---|---|---|---|
| Balaghat (40) | 1701698 | 1384 | 245 | 11 | 9 | 2 | 22.5% |
| Bhopal (27) | 2371061 | 519 | 1917 | 16 | 3 | 13 | 2.9% |
| Damoh (11) | 1264219 | 1210 | 251 | 8 | 6 | 2 | 13.2% |
| Dhar (21) | 2185793 | 1535 | 413 | 14 | 11 | 3 | 55.9% |
| Dindori (36) | 704524 | 924 | 32 | 5 | 5 | 0 | 64.7% |
| Hoshangabad (32) | 1241350 | 961 | 390 | 8 | 5 | 3 | 15.9% |
| Ratlam (17) | 1455069 | 1069 | 435 | 9 | 6 | 3 | 28.2% |
| Satna (12) | 2228935 | 1984 | 474 | 14 | 11 | 3 | 14.4% |
| Sheopur (01) | 687861 | 582 | 107 | 4 | 3 | 1 | 23.5% |
| Shivpuri (06) | 1726050 | 1417 | 295 | 11 | 9 | 2 | 13.2% |
| Total | 15566560 | 11585 | 4559 | 100 | 70 | 30 | 23.2% |

interval. The method was thus repeated till the desired sample size was achieved. Fig. S1 shows process of sampling and data collection.

## Participants and consent

All adults between 18–69 years of age who were residents of the selected cluster were eligible to be included in the study. Out of all the eligible members in household, respondent was selected as per methods already described above. Purpose of the study and procedures were explained to the respondent and a Participant Information Sheet (PIS) in local language was also provided. Subsequently those who provided written informed consent were included in the study.

## Study instrument / tool

We have used WHO -STEPS instrument for collection of data.[6][24] This instrument has got different risk factor assessment tool like socio-demographic information and behavioural risk factor assessment based on various question sets (STEP-1), physical measurement (STEP-2) and biochemical measurement (STEP-3). This questionnaire was translated in to Hindi to suit local needs before using for data collection. Copy of the study instrument is provided in supplementary files. Standard techniques were used for anthropometric measurements like height, weight, waist & hip circumference. Standard procedure mentioned in JNC 8 criteria were used to measure blood pressure of the individuals. For the purpose of measuring weight, we used Seca 803- Electronic Flat Weighing Scale, and for measuring the height we used Seca 213 Portable Stadiometer, both manufactured by Seca, Germany.

*Study variables*- Key study variables are described in Table 2. We have used operational definitions from WHO-STEPS manual (*World Health Organization (WHO), 2020b*).

*Study procedures*- Data collection was done by team of four field investigators and a laboratory technician who were trained in interview technique, anthropometry, blood pressure measurements and laboratory procedures. Quality checks were done by supervisor & study investigators by resurveying selected samples at each PSU to verify errors in data

**Table 2  Operational definitions used for the study.**

| Variable | Operational definition |
| --- | --- |
| Hypertension | Hypertension was defined as a mean systolic or diastolic blood pressure of ≥140 mmHg or ≥90 mmHg, respectively, in two serial measurements taken three minutes apart, or as current use of antihypertensive medication. |
| Impaired fasting glycaemia | Impaired fasting glycaemia was defined as a plasma venous value ≥ 6.1 mmol/L (110 mg/dl) and <7.0 mmol/L (126 mg/dl) or a capillary whole blood value ≥5.6 mmol/L (100 mg/dl) and <6.1mmol/L (110 mg/dl). |
| Raised fasting glucose | Raised fasting glucose was defined as a fasting glucose of ≥ 7.0 mmol/L (126 mg/dL) as measured in venous blood, or capillary whole blood value of fasting glucose of ≥ 6.1 mmol/L (110 mg/dl). |
| Raised total cholesterol | Raised total cholesterol was defined as a blood cholesterol level of ≥ 5.0 mmol/L or ≥ 190 mg/dl or currently on medication for raised cholesterol. |
| Obesity | Obesity was defined as a BMI equal to or greater than 30 kg/m$^2$ |
| Overweight | Overweight was defined as a BMI equal to or greater than 25 kg/m$^2$ |

collection. Accordingly, the field supervisor had to check 5 percent of interviews done by field workers in each cluster.

All data collection was done on a paperless data collection system, e-STEPS for android which was deployed on Ona platform. It had a STEPS app which was used for household member listing and respondent selection and e-STEPS app in which data collection forms for each step were available.

Data collection in each cluster was completed in 3–4 days. Field investigators administered Step-1 questionnaire and invited participant for physical measurement and laboratory testing on early morning next day. Individuals not available during the primary visit were visited once again. If they were found to be not available during the second visit also, they were classified as non-respondent.

*Biochemical investigations-* Fasting capillary glucose, total cholesterol and triglycerides tests were done by a trained laboratory technician by using point-of-care technology. For this we used Aina Mini Blood Glucose Monitoring System and Aina Lipids System of Jana Care Inc, Boston (MA).

**Ethics issues-** The study protocol was approved by Institutional Human Ethics Committee (IHEC) of AIIMS Bhopal. (Approval No IHEC-LOP/2014/EF0018 Dated 30th Jan 2015). Permission for data collection at field sites was obtained from Directorate Health Services, Government of Madhya Pradesh. All participants were explained about the study in Hindi and provided with participant information sheet. Interviews were conducted only after written informed consent was obtained. Individuals who were found to have high risk factors including raised blood pressure, blood glucose and total cholesterol were informed about it and referred to nearest public health facility for healthcare services.

## Data analysis

Data collected through mobile phone application was downloaded from server in the form of spreadsheet and was cleaned for any inconsistencies of implausible values. We have used IBM SPSS Statistics for Macintosh, Version 26.0. (Armonk, NY: IBM Corp.) for data analysis. Mean and standard deviation, or median and IQR (whichsoever was applicable) were used to summarize numerical variables as per their distribution. Nominal variables were summarized as frequency and percentage. Sampling weights for each STEP were derived by considering selection probabilities at different sampling stages. Subsequently weighted prevalence of behavioural and biological risk factors were calculated with their 95% confidence interval. Global NCD Monitoring Framework developed by WHO has recommended 25 indicators categorized under four broad domains *viz* mortality and morbidity, behavioural risk factors, biological risk factors and national health system response (*World Health Organization (WHO), 2020c*). Out of these 9 are estimated in results of this study.

## RESULTS

A total of 6000 eligible members from 100 clusters were approached for participation in study, of which 5680 consented. STEP-1 (socio-demographic and behavioural risk factors) data was collected for 5680 participants (STEP-1 response rate 94.6%), 4985 participants turned up for physical measurements (STEP-2 response rate- 83.1%) and laboratory testing was done for 4698 participants (STEP-3 response rate 78.3%). Mean age of participants was 40.4 years with standard deviation of 13.3 years. A total of 1,628 respondents (28.6%) belonged to the Scheduled Tribes category. Table 3 shows distribution of study participants as per age, gender and other socio-demographic characteristics.

### Prevalence of behavioural risk factors

Gender wise weighted prevalence of behavioural risk factors is shown in Table 4. One in four male and less than one percent females were found to be smoking tobacco daily. The mean age for starting smoking among both sexes was 22.18 years [95% CI [21.58–22.78]] and the mean number of manufactured cigarettes smoked per day was 0.28 [95% CI [0.11–0.44]]. About 5.7% of the daily smokers were reported of smoking manufactured cigarettes. Current alcohol consumption, i.e., alcohol consumption in anytime in the past 30 days was reported by 11.8% of males but only 0.3% of females. At the same time, 1.2% [95% CI [0.8–1.6]] professed to have engaged in heavy drinking in the last 30 days. The mean number of days fruit and vegetables were consumed by the respondents in a typical week were 1.36 days [95% CI [1.2–1.51]] and 5.09 days [95% CI [4.9–5.27]] respectively. The mean number of servings of fruit and vegetables that were consumed per day on an average was 0.82 [95% CI [0.76–0.88]] and 1.56 [95% CI [1.52–1.61]] respectively. Majority (98.5% [95% CI [97.9–98.9]]) of the respondents ate less than 5 servings of fruit and/or vegetables on an average per day, and about one fourth (25.5% [95% CI [22.8–28.4]]) of the respondents always or often added salt or salty sauce to their food before eating or as they were eating. 11.9% [95% CI [10.1–13.9]] always or frequently ate processed foods high in salt.

**Table 3  Distribution of participants by various Socio-demographic characteristics.**

| Variables | Both sexes | Men | Women |
| --- | --- | --- | --- |
| Mean Age (SD) | 40.4 (13.6) | 41.5 (13.9) | 39.7 (13.3) |
| Age Group | | | |
| 18–29 | 1377 (24.2) | 452 (21.9) | 925 (25.5) |
| 30–44 | 2095 (36.9) | 744 (36) | 1351 (37.3) |
| 45–59 | 1459 (25.7) | 559 (27) | 899 (24.9) |
| 60–69 | 739 (13) | 308 (14.9) | 439 (12.1) |
| Mean Years of Education (SD) | 4.6 (4.8) | 5.9 (4.9) | 3.8 (4.7) |
| Median Annual Per Capita Income (INR) (IQR) | 3000 (9000) | 3333.3 (9000) | 3000 (7750) |
| Marital Status (%) | | | |
| Never Married | 386 (6.8) | 230 (11.1) | 156 (4.3) |
| Currently married | 4829 (85.1) | 1760 (85.3) | 3069 (84.9) |
| Divorced or Widowed | 462 (8.1) | 73 (3.5) | 389 (10.8) |
| Employment Status (%) | | | |
| Employed | 5384 (94.9) | 1918 (93) | 3466 (95.9) |
| Unemployed or Students | 293 (5.2) | 144 (7) | 149 (4) |
| Tribal Group (%) | | | |
| Scheduled Tribe | 1628 (28.6) | 611 (29.6) | 1017 (28.1) |
| Other | 4049 (71.3) | 1453 (70.4) | 2599 (71.9) |

## Prevalence of biologic risk factors

The mean BMI of respondents was 20.88 kg/m$^2$. About 15.3% [95% CI [12.8–18.2]] of the respondents were overweight and 3.4% [95% CI [2.6–4.3]] were obese. More females were found to be overweight and obese as compared to males. The mean systolic and diastolic blood pressure of the respondents was 125.94 [95% CI [124.98–126.89]] and 77.68 [95% CI [77.09–78.27]] mm Hg respectively. Almost a fourth (22.3% [95% CI [20.5–24.1]]) of the respondents had raised BP or were currently on medication for raised BP. The mean fasting blood glucose of the respondents including those currently on medication for raised blood glucose was 92.69 [95% CI [90.2–95.18]] mg/dl. Prevalence of raised fasting glucose was 6.8% (95% CI [5.6–8.3]) which was higher in males (8.4% [95% CI [6.6–10.7]]).

## Summary of risk factors

We estimated the percentage of participants who had 3 or more risk factors and stratified it by age (below 45 years and above 45 years) and gender (male and female). Men were found to be having more multiple risk factors in both the age categories (36.6 and 54.6 percent) as compared to females. Within females, burden of multiple risk factors increases sharply from 17.0 percent below 45 years to 38.1 percent after 45 years.

The alluvial diagram (Fig. 2) attempts to connect the demographic characteristics (age and gender) to behavioural, anthropological and biological NCD risk magnitude in cohesion. There seems to be some distinctive patterns here- first, age and behavioural risk factors show an incremental relationship which is more lucid in male. Second, females are more prone to 'being overweight' in spite of lesser propensity to affect with behavioural risk . This rivulet of alluvial diagram can be seen translated into biological risk for NCD.

**Table 4 Prevalence of behavioural and biologic risk factors (in percent) stratified by gender.** Prevalence is shown in percent and 95% confidence interval is presented in parenthesis.

| Results for adults aged 18–69 years in *percent* (incl. 95 CI) | Males | Females | Both sexes |
|---|---|---|---|
| **Step 1 Tobacco Use** | | | |
| Percentage who are current tobacco users (any form) | 61.0 (57.5–64.4) | 18.8 (15.8–22.1) | 34.2 (31.5–36.9) |
| Percentage who currently smoke tobacco | 24.3 (21.0–27.9) | 0.8 (0.5–1.2) | 9.4 (7.9–11.0) |
| Percentage who currently smoke tobacco daily | 20.9 (17.8–24.4) | 0.7 (0.4–1.1) | 8.1 (6.7–9.6) |
| *For those who smoke tobacco daily* | | | |
| Average age started smoking (years) | 21.93 (21.35–22.51) | 26.71 (23.03–30.39) | 22.18 (21.58-22.78) |
| Percentage of daily smokers smoking manufactured cigarettes | 5.8 (3.4–9.6) | 4.3 (0.6–24.6) | 5.7 (3.4–9.3) |
| Mean number of manufactured cigarettes smoked per day (by smokers of manufactured cigarettes) | 0.28 (0.11–0.46) | 0.17 (0.0–0.50) | 0.28 (0.1–0.44) |
| **Step 1 Alcohol Consumption** | | | |
| Percentage who are lifetime abstainers | 82.8 (80.0–85.3) | 99.3 (99.0–99.6) | 93.3 (92.1–94.3) |
| Percentage who are abstainers for the past 12 months | 84.9 (82.1–82.3) | 99.6 (99.3–99.8) | 94.2 (93.1–95.2) |
| Percentage who currently drink (drank alcohol in the past 30 days) | 11.8 (9.7–14.2) | 0.3 (0.2–0.6) | 4.5 (3.7–5.5) |
| Percentage who engage in heavy episodic drinking (6 or more drinks on any occasion in the past 30 days) | 3.1 (2.2–4.3) | 0.1 (0.0–0.2) | 1.2 (0.8–1.6) |
| **Step 1 Diet** | | | |
| Mean number of days fruit consumed in a typical week | 1.4 (1.24–1.56) | 1.34 (1.16–1.51) | 1.36 (1.2-1.51) |
| Mean number of servings of fruit consumed on average per day | 0.85 (0.79–0.91) | 0.8 (0.73–0.88) | 0.82 (0.76–0.88) |
| Mean number of days vegetables consumed in a typical week | 5.04 (4.82–5.25) | 5.12 (4.92–5.31) | 5.09 (4.9–5.27) |
| Mean number of servings of vegetables consumed on average per day | 1.54 (1.48–1.59) | 1.58 (1.53–1.63) | 1.56 (1.52–1.61) |
| Percentage who ate less than 5 servings of fruit and/or vegetables on average per day | 98.5 (97.7–99.0) | 98.5 (97.9–98.9) | 98.5 (97.9-98.9) |
| Percentage who always or often add salt or salty sauce to their food before eating or as they are eating | 28.7 (24.9–32.8) | 23.7 (21.0–26.5) | 25.5 (22.8–28.4) |
| Percentage who always or often eat processed foods high in salt | 12.4 (10.2–14.8) | 11.6 (9.7–13.8) | 11.9 (10.1–13.9) |
| **Step 1 Physical Activity** | | | |
| Percentage with insufficient physical activity (defined as <150 min of moderate-intensity activity per week, or equivalent) | 25.3 (21.7–29.2) | 16.3 (13.4 –19.7) | 19.6 (17.0- 22.5) |
| **Results for adults aged 18–69 years (incl. 95 CI)** | **Males** | **Females** | **Both Sexes** |
| **Step 1 Cervical Cancer Screening** | | | |
| Percentage of women aged 30-49 years who have ever had a screening test for cervical cancer | | 1.5 (1.0–2.5) | |
| **Step 2 Physical Measurements** | | | |
| Mean body mass index - BMI (kg/m$^2$) | 20.4 (20.17–20.78) | 21.1 (20.69–21.54) | 20.89 (20.54–21.23) |
| Percentage who are overweight (BMI $\geq$ 25 kg/m$^2$) | 10.6 (8.4—13.4) | 18.0 (14.8–23.6) | 15.3 (12.8–18.2) |
| Percentage who are obese (BMI $\geq$ 30 kg/m$^2$) | 1.6 (1.1–2.3) | 4.4 (3.3–5.7) | 3.4 (2.6-4.3) |
| Average waist circumference (cm) | 79.99 (78.76–81.2) | 77.39 (75.99–78.8) | |

**Table 4** (*continued*)

| Results for adults aged 18–69 years in *percent* (incl. 95 CI) | Males | Females | Both sexes |
| --- | --- | --- | --- |
| Mean systolic blood pressure - SBP (mmHg), including those currently on medication for raised BP | 127.87 (126.7–129.04) | 124.89 (123.7–126.01) | 125.94 (124.98-126.89) |
| Mean diastolic blood pressure - DBP (mmHg), including those currently on medication for raised BP | 78.74 (77.94–79.54) | 77.1 (76.45–77.76) | 77.68 (77.09–78.27) |
| Percentage with raised BP (SBP $\geq$140 and/or DBP $\geq$90 mmHg or currently on medication for raised BP) | 23.5 (20.7–26.4) | 12.5 (9.5–16.4) | 22.3 (20.5–24.1) |
| Percentage with raised BP (SBP $\geq$ 140 and/or DBP $\geq$ 90 mmHg) who are not currently on medication for raised BP | 10.6 (7.6–14.6) | 12.5 (9.5–16.4) | 11.8 (9.3–14.9) |
| **Step 3 Biochemical Measurement** | | | |
| Mean fasting blood glucose, including those currently on medication for raised blood glucose [choose accordingly: mmol/L or mg/dl] | 96.99 (93.88–100.10) | 90.4 (87.94–92.86) | 92.69 (90.2–95.18) |
| Percentage with impaired fasting glycaemia as defined below | | | |
| • plasma venous value $\geq$6.1 mmol/L (110 mg/dl) and <7.0mmol/L (126 mg/dl) | 28.6 (23.0–34.8) | 19.9 (16.0–24.5) | 22.9 (18.7–27.8) |
| • capillary whole blood value $\geq$5.6 mmol/L (100 mg/dl) and <6.1 mmol/L (110 mg/dl) | | | |
| Percentage with raised fasting blood glucose as defined below or currently on medication for raised blood glucose | | | |
| • plasma venous value $\geq$ 7.0 mmol/L (126 mg/dl) | 8.4 (6.6–10.7) | 6.0 (4.7–7.5) | 6.8 (5.6–8.3) |
| • capillary whole blood value $\geq$ 6.1 mmol/L (110 mg/dl) | | | |
| Mean total blood cholesterol, including those currently on medication for raised cholesterol [choose accordingly: mmol/L or mg/dl] | 119.54 (117.14–121.93) | 128.08 (125.65–130.51) | 125.12 (122.95-127.29) |
| Percentage with raised total cholesterol ( $\geq$ 5.0 mmol/L or $\geq$ 190 mg/dl or currently on medication for raised cholesterol) | 3.1 (2.3–4.2) | 4.9 (3.9–6.2) | 4.3 (3.4–5.3) |
| **Summary of combined risk factors** | | | |
| • current daily smokers | | | |
| • less than 5 servings of fruits & vegetables per day | | | |
| • insufficient physical activity | | | |
| • overweight (BMI $\geq$ 25 kg/m$^2$) | | | |
| • raised BP (SBP $\geq$ 140 and/or DBP $\geq$ 90 mmHg or currently on medication for raised BP) | | | |
| Percentage with none of the above risk factors | 0.1 (0–0.4) | 0.4 (0.2–0.8) | 0.3 (0.2–0.6) |
| Percentage with three or more of the above risk factors aged 18 to 44 years | 36.6 (33.4–39.8) | 17.0 (14.9–19.5) | 24.2 (22.1–26.3) |
| Percentage with three or more of the above risk factors aged 45 to 69 years | 54.6 (50.0–59.0) | 38.1 (34.7–41.6) | 44.0 (41.1–46.9) |
| Percentage with three or more of the above risk factors aged 18 to 69 years | 41.6 (38.7–44.5) | 23.1 (20.8–25.4) | 29.8 (27.7–31.9) |

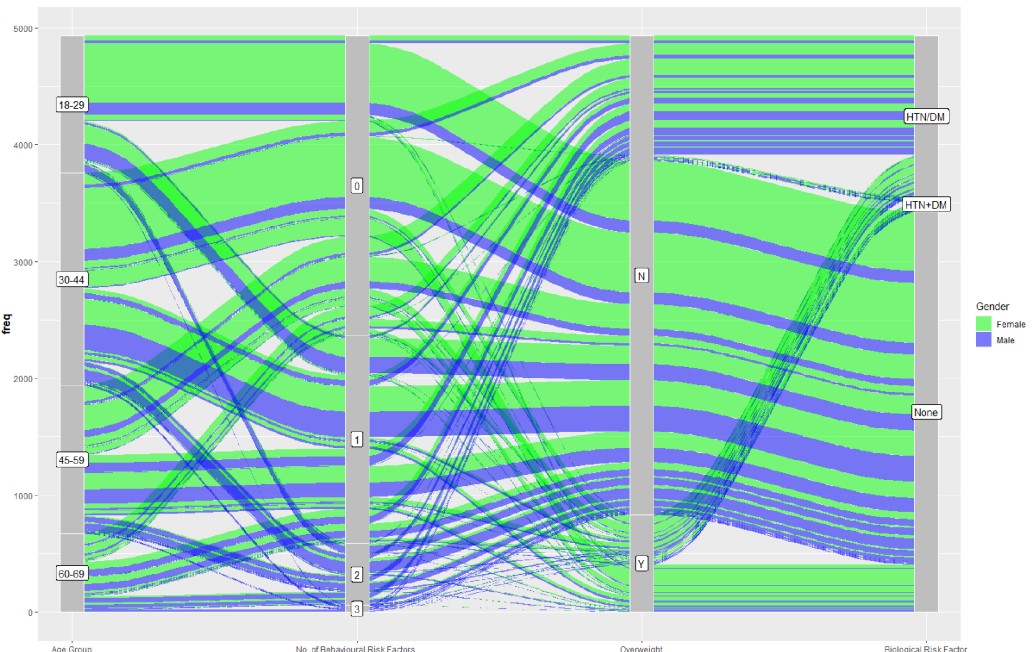

**Figure 2** **Demographic trajectories in reference to behavioural and biological risk magnitude.** Pillars in the alluvial diagram shows demographic characteristics (age group), number of behavioural risk factors, presence of overweight and presence of biological NCD risk factor. Colours of rivulet represent gender and its size represent number of participants.

Another interesting observation is related to biological risk which shows a pattern seemingly disfavorable to male.

## DISCUSSION

In this cross-sectional study representative of Madhya Pradesh, we found that 9.4% individuals smoked tobacco, 15.3% were overweight and obese, 22.3% had hypertension, and 6.8% had diabetes mellitus. The risk factors were seen to have important gender-based differences. As compared to women, men were less likely to be overweight or obese, but more likely to smoke tobacco, and have diabetes mellitus. Hypertension was also more common in men. Overall about a fourth of all adults had three or more risk factors for cardiovascular disease.

A systematic review of all large NCD risk factor surveys conducted either nationally or intra-state identified three national surveys, two surveys with representation from multiple states, and two state wide surveys from Punjab and Kerala (*Nethan, Sinha & Mehrotra, 2017*). Subsequently a third State-wide survey was reported from Haryana (*Thakur et al., 2019*). All the previous state-wide surveys are from high or high-middle ETL states. The current study from Madhya Pradesh (MP), the fourth state-wide survey (after Punjab, Kerala, and Haryana) and the first from a low ETL state adds to the evidence on burden of NCDs across various parts of the country. Previous National IDSP-NCD Survey (2007-08), that included Madhya Pradesh (*National Institute of Medical Statistics ICMR, 2009*) had

about 5000 respondents from the State. In a decadal comparison to this survey, smoking prevalence in MP seems to have reduced drastically (22% in IDSP-NCD Survey vs 9.4%), though prevalence of hypertension shows mild reduction (24% vs 22.3%). The trend for tobacco consumption is reflected in the NFHS-4 data too, wherein the prevalence of tobacco consumption is found to be 16.9%, i.e., in between our finding and findings a decade ago, indicating a downwards secular trend. On the other hand, prevalence of hypertension was found to be lower in NFHS-4 (10.1%), as compared to our findings (*NFHS4 MP, 2020*). The declining trend in smoking seems to be pan-India, as it has also been documented in various NCD surveys conducted over the last two decades, with an overall national decline from 34.6% in 2004-05 to 28.6% in 2016-17 (*Nethan, Sinha & Mehrotra, 2017*). At the same time, a systematic review of national surveys has also documented a rise in prevalence of being obese or overweight in both genders. A nationwide NCD monitoring survey was recently concluded in 2017-18, results of which are awaited. These results taken together can provide valuable clues about the secular trend of NCD burden in the country and the risk factors associated with them, thereby helping to shape NCD policy and strategy by assessing the capacity health system to respond, and in advocacy to the policy makers (*Narain & Thankappan, 2018*).

When comparing the State wide surveys from Punjab (PB) (*Thakur et al., 2016*), Haryana (HR) (*Thakur et al., 2019*), and Kerala (KL) (*Sarma et al., 2019*), it can be seen that the prevalence of smoking tobacco is highest in Haryana (25.5% in HR, 11.6% in PB, 9.4% in MP, vs 7.2% in KL). On the other hand, prevalence of being overweight or obese is lowest in MP (40.4% in PB, 38.3% in KL, 35.1% in HR, vs 15.3% in MP). Prevalence of hypertension is highest in Punjab and lowest in Haryana (26.2% in HR, 22.3% in MP, 30.4% in KL, vs 40.1% in PB). Prevalence of diabetes is also higher (19.2%) in Kerala, as compared to Madhya Pradesh (6.8%). Average fasting blood sugars are lowest in Madhya Pradesh (MP 92.69 mg/dL; HR 97.5 mg/dL; PB 96.9 mg/dL; KL 108.6 mg/dL), while average systolic blood pressures are also on the lower side (HR 123.7 mm Hg; MP 125.94 mm Hg; KL 126.6 mm Hg; PB 130.4 mm Hg). These regional differences have pathophysiological underpinnings, and are also important to plan and develop strategies to control NCDs.

These percentages, when converted to sheer numbers bring into focus the burden of NCDs which is faced by the state health machinery. Census 2011 pegged the population of Madhya Pradesh at 72.62 million. Taking this to be the base population at the time of the survey, it can be estimated 6.8 million across the state smoke tobacco products. At the same time, 11.1 million are overweight or obese, 16.2 million are hypertensive, while 4.9 million are diabetic.

Various local surveys have also been reported in restricted population subsets from India in previous five years between 2014 and 2019 (Table 5). These surveys often focus on population subsets that may be more vulnerable to risk factors. It is likely that local prevalence estimates are different from State-wide estimates. Many local surveys have exclusively been performed in tribal pockets, where prevalence of some risk factors such as smoking is particularly high.

The findings from the current study is strengthened by the systematic sampling for participants, which is representative of MP as a whole. Limitation is the cross-sectional

**Table 5  Summary of Local/Regional WHO-STEPS community-based CVD risk factorassessment from India.**

| Author; Year | Location Number screened | Tobacco consumption (form of tobacco) | Overweight and Obese prevalence | HTN Prevalence Mean SBP | DM Prevalence Mean BS |
|---|---|---|---|---|---|
| *Bhardwaj et al. (2014)* | Hamirpur (Himachal Pradesh) $n = 2749$ | NR | 31.3% | 37.4% 131.5 mm Hg | NR |
| *Bhar, Bhattacherjee & Das (2019)* | Siliguri (West Bengal) $n = 172$ | 25.6% (smoking) | 12.2% | NR | NR |
| *Shikha et al. (2019)* | Garhwal (Uttarakhand) $N = 632$ | | 33.3% | | |
| *Ramaswamy et al. (2019)* | Tamil Nadu $N = 873$ | 7.2% (smoking) | 48% | | 2.2% |
| *Singh et al. (2017)* | Chittor (Andhra Pradesh) $N = 16636$ | 8.4% (smoking) | 29% | 21.6% 124 mm Hg | 6.9% |
| *Sathish et al. (2017)* | Thiruvananthapuram (Kerala) $N = 410$ | 15.3% (smoking) | 58% | 43.2% 134.8 mm Hg | |
| *Bhattacherjee et al. (2015)* | Siliguri (West Bengal) $n = 779$ | 57.5% (all tobacco use) | 29.8% (overweight) 20.2% (abd obesity) | 17.8% | 9.1% |
| *Misra, Mini & Thankappan (2014)* | Tinsukia (Assam) $n = 332$ | 84% (all tobacco use) | 16% (overweight) 11% (abd obesity) | 26% | |
| *Sajeev & Soman (2018)* | Thiruvananthapuram (Kerala) $n = 298$ | 81.5% (all tobacco use) | 22.1% (abd obesity) | 48.3% | |
| *Harikrishnan et al. (2018)* | Kerala –three districts $n = 5063$ | 17.8% (all tobacco use) | 30% (overweight) 8.7% (obesity) 27.0% (abd obesity) | 28.9% | 15.6% |
| *Srivastav et al. (2017)* | Gautam Buddha Nagar (Uttar Pradesh) $n = 207$ | 12.6% (smoking) 22.7% (smokeless tobacco) | 26.6% (overweight) 6.8% (obesity) | 18.4% | 9.7% |
| *Kandpal, Sachdeva & Saraswathy (2016)* | Uttarakhand (Uttarakhand) $n = 288$ | 13.9% (smoking) | 56.6% (overweight and obesity) 33.7% (abd obesity) | 43.4% | 6.9% |
| *Kumar, Choudhury & Yadav (2018)* | Lefunga Block (Tripura) $n = 150$ | 26% (smoking) 68% (smokeless tobacco) | 26% (overweight) 45.3% (abd obesity) | 31% | |
| *Tushi et al. (2018)* | Mokokchung district (Nagaland) $n = 472$ | 19.5% (smoking) 54.2% (smokeless tobacco) | 32.4% (overweight) 8.8% (obesity) 34.8% (abd obesity) | 43.2% | |
| *Deo et al. (2018)* | Maharashtra –4 districts $n = 1864$ | 46.5% (all tobacco use) | 0.9% (obesity) | 11.7% | 6.7% |

design, which is not suitable to temporally associate the risk factors with development of NCDs. Further, like all surveys there is likely to be a desirability bias in the respondents, that could have underestimated prevalence of behavioural risk factors like smoking. This study had a greater sampling of individuals who would be at home at the time of survey reflected in preponderance of women in the sample. Thus, the burden of gender discriminant risk factors is likely to be biased.

High NCD burden in India, and a high contribution of DALYs due to them is a concern. In addition there are huge regional disparities (*Arokiasamy, 2018*) calling for regional variations in NCD control action plans. While one size may not fit all, state-wide and local modifications in NCD action plan would be needed (*Jayanna et al., 2019*). Further we need to move from isolated surveys, to intermittent surveillance strategies. These strategies will be useful to document success of complex individual, societal, environmental and administrative strategies required for risk-factor control.

## CONCLUSION

The survey shows that a large section of the population from Madhya Pradesh is either suffering from NCDs or have risk factors which predispose them to acquire NCDs. This state representative survey provides benchmarking information for behavioural and biological risk factor distribution for recently scaled up National Programme for the Prevention and Control of Cancer, Diabetes, Cardiovascular Diseases, and Stroke (NPCDCS).

**Abbreviations**

| | |
|---|---|
| **BMI** | Body Mass Index |
| **CEB** | Census enumeration blocks |
| **CI** | Confidence Interval |
| **CMNNDs** | Communicable, Maternal, Neonatal, and Nutritional Diseases |
| **DALYs** | Disability Adjusted Life Years |
| **DBP** | Diastolic Blood Pressure |
| **EAG** | Empowered Action Group |
| **ETL** | Epidemiological Transition Level |
| **FBG** | Fasting Blood Glucose |
| **GBD** | Global Burden of Disease |
| **HR** | Haryana |
| **IDSP** | Integrated Disease Surveillance Programme |
| **IMR** | Infant Mortality Rate |
| **JNC** | Joint National Committee |
| **KL** | Kerala |
| **MP** | Madhya Pradesh |
| **MMR** | Maternal Mortality Rate |
| **NCDs** | Non-communicable diseases |
| **NPCDCS** | National Programme for the Prevention and Control of Cancer, Diabetes, Cardiovascular Diseases, and Stroke |
| **NR** | Not Reported |
| **PB** | Punjab |

| **PIS** | Participant Information Sheet |
|---|---|
| **PSU** | Primary Sampling Units |
| **SBP** | Systolic Blood Pressure |
| **ST** | Scheduled Tribes |
| **STEPs** | STEPwise approach to Surveillance |
| **WHO** | World Health Organization |

### Funding
Indian Council of Medical Research, New Delhi (ICMR) funded this study under extramural grants. (IRIS No 2014-2668). The funders had no role in study design, data collection and analysis, decision to publish, or preparation of the manuscript.

### Grant Disclosures
The following grant information was disclosed by the authors:
Indian Council of Medical Research, New Delhi (ICMR): IRIS No 2014-2668.

### Competing Interests
Abhijit P. Pakhare is an Academic Editor for PeerJ. The remaining authors declare that they have no competing interests.

### Author Contributions
- Arun M. Kokane conceived and designed the experiments, performed the experiments, analyzed the data, authored or reviewed drafts of the paper, and approved the final draft.
- Rajnish Joshi and Ashwin Kotnis conceived and designed the experiments, performed the experiments, authored or reviewed drafts of the paper, and approved the final draft.
- Anirban Chatterjee analyzed the data, authored or reviewed drafts of the paper, and approved the final draft.
- Kriti Yadav, G Revadi and Ankur Joshi analyzed the data, prepared figures and/or tables, and approved the final draft.
- Abhijit P. Pakhare conceived and designed the experiments, performed the experiments, analyzed the data, prepared figures and/or tables, authored or reviewed drafts of the paper, and approved the final draft.

### Human Ethics
The following information was supplied relating to ethical approvals (i.e., approving body and any reference numbers):
Institutional Human Ethics Committee (IHEC) of AIIMS Bhopal, India, approved this study (IHEC-LOP/2014/EF0018 Dated 30th Jan 2015).

### Field Study Permissions
The following information was supplied relating to field study approvals (i.e., approving body and any reference numbers):
Permission and facilitation for data collection at field sites were provided by Directorate Health Services, Satpura Bhavan, Madhya Pradesh.

## Data Availability

Raw data with standard variable names is available as a Supplemental File.

## Supplemental Information

Supplemental information for this article can be found online at http://dx.doi.org/10.7717/peerj.9568#supplemental-information.

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
