# Peer review of "Descriptive profile of risk factors for cardiovascular diseases using WHO STEP wise approach in Madhya Pradesh"

_PeerJ, doi:10.7717/peerj.9568_

## Round 0.1 · original submission · Major Revisions

I have read the manuscript and the reviews. I do agree that the manuscript needs improvement, and the authors should address the reviewers' comments especially reviewer 3 in revised submission.

·

Basic reporting

The manuscript uses a standard technique (WHO) and questionnaire for the conduct of the study, However I would like to see a few things added:
1. A flow chart describing the methodology.
2. A comparison with the NFHS data with regard to behavioral risk factors.

Experimental design

No comment

Validity of the findings

No Comment

Additional comments

No Comment

Reviewer 2 ·

Basic reporting

The manuscript is well written, clearly articulated, and self contained with relevant results for the stated hypothesis. There are a few minor edits that I have outlined below for the authors to clarify/rectify -
- Line 41 - DALYs- expand DALYs
- Line 64 - 'clusters' to 'cluster'
- Line 73 - This sentence is unclear. Suggest to rephrase this sentence for clarity.
- Line 78 - This sentence is unclear. Suggest to rephrase for clarity.
- Line 88 - expand KISH and add a reference
- Line 89 - This sentence is unclear. Suggest to rephrase for clarity.
- Line 95 - Sentence is incomplete
- Line 100 - The make and manufacturer of the instruments used to measure height, weight and blood pressure need to be mentioned.
- Line 108 - It is unclear if a resurvey involved a systematic resampling of the sampled individuals or it was random. Also, it is unclear if there is a fixed number per cluster that was resampled. Would be good to mention this in the manuscript.
- Line 115 - The sentence needs to be corrected for grammar.

Experimental design

No comment

Validity of the findings

No comment

Additional comments

No comments

Reviewer 3 ·

Basic reporting

The abstract seems quite lengthy. The authors may trim it to retain only the vital information/summary of the project.

Line 41: What is DALY?

Line 95: The sentence seems to end abruptly.

Table 3: The authors may add info about total men/women participating at each stage of the survey. For Per capita income, the SD is quite high – this is perhaps due to the presence of data from rural & urban centres in this study. Maybe the authors can present the data for rural/urban separately to have more focussed stats.

Figure 2: The data has been represented by innovative means, but the figure legend does not clearly provide all the information. The authors must ensure that the legends are more detailed. Similarly, Table 5 has some abbreviations which are not explained in the legends or in another part of manuscript (for example, TR, AS,..)

The authors must also ensure the use of standard terms throughout the paper – Hypertension is mentioned as HTN in the tables and HT in the figures. It would be better to use a single identifier.

Line 194: Fourth or third from Haryana? If so, which is the other state? The statement is not very clear. Kindly clarify. It contradicts with Line 196.

Line 227: Table 4 or Table 5? Table 5 has not been cited inline in the manuscript

Experimental design

The study design has been planned well, in accordance to the WHO-STEPS protocols and it is laudable that the data has been collected in a paper-free manner, eliminating a lot of possible errors during the collection and consolidation phases.

Validity of the findings

While the study design is good, the authors could try to make the maximum use of the collected data to check on the trends in NCDs in age-group related manner as well.

At present, Table 4 provides data pertaining to the total population, as well as male/female numbers. It would be interesting to see how the trends are in each age group – whether there is a significant burden in a particular age-group for a particular trait, so to speak. The authors may even choose to provide this data in the supplementary section.

The authors mention that the smoking index has dropped drastically in MP since the data from 10 years ago – however, the previous study was a state-wide survey, as against the current study in only 10 districts, which only 30% representation in the Urban sector – perhaps, this might be a part of the reason for the drop in numbers?

Additional comments

This manuscript by Kokane and colleagues outlines the risk factors for cardiovascular diseases in the state of Madhya Pradesh in India, using WHO STEP guidelines. The work done is extensive and well presented. However, there are a few parameters which the authors may further fine-tune as suggested here.

The manuscript would benefit from being reviewed completely by a native English speaker or a language editor. There are multiple instances where there is a lack of connectivity/flow and coherence.

The table headings seem to indicate all tables as Table-1: the authors may kindly fix this issue (refer pages 25,27,29,32…)

Table 2 may be provided as supplementary material.

---

## Round 0.2 · accepted · Accept

I am delighted to report that the concluding remarks received from all Reviewers are positive. I appreciate the time and effort you have undertaken to address the Reviewer’s comments. The manuscript has been revised as per the reviewer’s comments and now deemed fit for publication in PeerJ. Congratulations